# A self-sustainable wearable multi-modular E-textile bioenergy microgrid system

Lu Yin[1,2], Kyeong Nam Kim[1,2], Jian Lv[1,2], Farshad Tehrani[1], Muyang Lin[1], Zuzeng Lin[1], Jong-Min Moon[1], Jessica Ma[1], Jialu Yu[1], Sheng Xu [1] & Joseph Wang [1✉]

Despite the fast development of various energy harvesting and storage devices, their judicious integration into efficient, autonomous, and sustainable wearable systems has not been widely explored. Here, we introduce the concept and design principles of e-textile microgrids by demonstrating a multi-module bioenergy microgrid system. Unlike earlier hybrid wearable systems, the presented e-textile microgrid relies solely on human activity to work synergistically, harvesting biochemical and biomechanical energy using sweat-based biofuel cells and triboelectric generators, and regulating the harvested energy via supercapacitors for high-power output. Through energy budgeting, the e-textile system can efficiently power liquid crystal displays continuously or a sweat sensor-electrochromic display system in pulsed sessions, with half the booting time and triple the runtime in a 10-min exercise session. Implementing "compatible form factors, commensurate performance, and complementary functionality" design principles, the flexible, textile-based bioenergy microgrid offers attractive prospects for the design and operation of efficient, sustainable, and autonomous wearable systems.

[1] Department of Nanoengineering, Center of Wearable Sensors, University of California San Diego, La Jolla, CA, USA. [2]These authors contributed equally: Lu Yin, Kyeong Nam Kim. ✉email: josephwang@ucsd.edu

The rapid rise of flexible electronics brings forth a myriad of sensors, circuits and energy storage devices in various wearable form factors[1–9]. In order to meet the growing power demands of wearable electronics and eliminate the need for frequent, interrupting recharges and cumbersome wired power transmission, wearable systems have integrated energy harvesters such as solar cells, triboelectric generators (TEGs), and enzymatic or microbial biofuel cells (BFCs) to enable their self-sustainable operation[10–16]. Different wearable devices have recently adapted this strategy to collect energy from human or the environment followed by regulating and storing the scavenged energy in storage modules such as batteries or supercapacitors (SCs)[17–23]. However, the operation of these systems has relied on either a monolithic input source which shares the same limitation in energy availability (e.g., the lack of motion, biofuel, sunlight), or upon multiple harvesters that operate in parallel but are not synergistic in realistic scenarios, and introduce additional limitations instead of compensating existing ones[24–29]. Moreover, the early multi-input hybrid harvesting systems, such as the integration of solar cells with SCs and batteries, relied partially on energy inputs from the external environment (e.g., thermoelectric, pyroelectric, photovoltaic) which are often uncontrollable[24–27,30]. To scavenge energy efficiently and reliably solely from human activities, system-level considerations are urgently needed to guide the judicious selection of components with complementary characteristics and commensurate performance.

In this regard, wearable energy systems can seek inspiration in the design and deployment of microgrids operating in "island mode"[31–33]. Microgrid, namely, a micro-scale power grid with components including energy generation, energy storage, various utilities, and management functionality for regulating the flow of energy, can be made self-sustainable and independent from the main power grid by the inclusion of various renewable energy harvesters and appropriate energy storage units. Such self-sustainable microgrids can operate independently from the main grid by harvesting energy from localized sources and regulating and storing the scavenged energy in various energy storage modules. The reliability of renewable sources is fortified by coupling with other generation sources (e.g., fuel-based generators) to ensure a timely energy supply[33–37]. Furthermore, these localized sources are paired with storage modules with optimal capacities based on the load energy demands[38]. Beyond the simple addition of harvesting and storage modules, design of the microgrid relies on the careful selection of components with compatible performance and complementary characteristics.

Inspired by this notion, we herein propose and demonstrate the concept of a wearable e-textile microgrid system: a multi-module, textile-base system with applications powered by complementary and synergistic energy harvesters and commensurate energy storage modules. To demonstrate this concept, we describe in the following section an integrated e-textile microgrid system that unite BFCs and TEGs, two harvesters with distinct and complementary energy conversion mechanisms based on human activities, along with SC modules for regulating the powering of wearable applications with both low and high power demand (Fig. 1a). Among many proposed integrated harvesters, the pairing of the biomechanical and biochemical harvesters is desirable as they rely solely on human activities. Previous studies demonstrated only such combination working in proof-of-concept in-vitro settings, hence are challenging to be employed for real-life scenarios[39–41]. Adapting the microgrid design concept, this work seamlessly integrates biomechanical and biochemical harvesters along with energy storage devices, with carefully budgeted energy rating, into one e-textile platform. When this microgrid harvests energy during human movements, the TEG storage modules are firstly activated from the instant motion-induced charge generation to harvest biomechanical

energy to rapidly boot the system, while the subsequently activated BFCs harvest biochemical energy from electroenzymatic reactions of sweat metabolites for prolonged power delivery (Fig. 1b). The complementary relationship between the two bioenergy harvesters thus compensates for the limitations of the BFCs due to delayed perspiration and of the TEGs due to the lack of motion. The SC modules regulate low-current, high voltage inputs from the TEG modules and high-current, low-voltage inputs from the BFC modules, with optimal capacity to delivery sufficient power for designated applications while maintaining fast booting. The optimized system can thus boot quickly within 3 min to continuously power a microwatt-rated wristwatch with liquid crystal display (LCD), or a milliwatt-rated sensor-electrochromic display (ECD) system operating in pulsed sessions, and extend their operation to over 30 min in connection to a 10-min movement session (Fig. 1c). Compared to the early integrated wearable energy systems, the present system relies solely on energy inputs from human activities and hence is not dependent on the external environment[24,28,29]. The performance of the individual modules was characterized, and the energy carefully budgeted to ensure that the limited amount of harvested energy was efficiently utilized. For compatibility with the wearable form factor, all modules are printed, textile-based, durable and flexible, and can be readily integrated onto a shirt to harvest energy from the sliding motion between the forearms and the torso by TEGs and the sweat generated above the chest by the BFCs (Fig. 1d). These modules are connected with flexible printed silver interconnections that are secured and insulated by a hydrophobic, water-proof polystyrene-polyethylene-polybutylene-polystyrene (SEBS) block copolymer. The placement of the modules ensures the optimal collection of biomechanical energy via the TEG module from the arm movements, along with an intimate contact of BFCs to the skin for sweat collection. The placement of BFCs and SCs on the chest also minimizes the possible bending and wrinkling deformation which may affect their energy harvesting and collection efficiency. Implementing the "complementary, commensurate, compatible" design principles, the microgrid e-textile system serves as an attractive example for future integrated on-body systems that are autonomous, reliable, synergistic, sustainable and energy-efficient. While the microgrid concept is introduced here to the field of wearable electronics using complementary BFC and TEG bioenergy harvesters paired with SC storage modules, it can be applied for guiding the development of future miniaturized energy systems based on a judicious selection and integration of different modules toward a variety of self-powered electronics applications.

## Results

**Characterization of the microgrid modules**. TEGs have been selected as the biomechanical energy harvesters in this wearable microgrid system due to their instant response to motions to generate energy. Since their inception in 2012, TEGs have become the most studied wearable mechanical energy harvesters due to its simple generation mechanism and the abundant selection of materials[10,42]. The present TEG module is composed of a polytetrafluoroethylene (PTFE)-based negative mover and an ethylcellulose-polyurethane (EC-PU)-based positive stator in interdigitated patterns (Fig. 2a and Supplementary Fig. 1), and is designed to harvest energy from sliding motions as illustrated in Fig. 2b[43–45]. To ensure the compatibility of the fabrication of the TEG elements to the e-textile platform with abundant flexibility and durability, each individual layer in the TEG was formulated into a screen-printable ink with elastomeric binder that is water-proof and resistant to abrasion. The TEG modules were characterized at low sliding frequencies ranging from 0.833 Hz to 3 Hz, simulating the

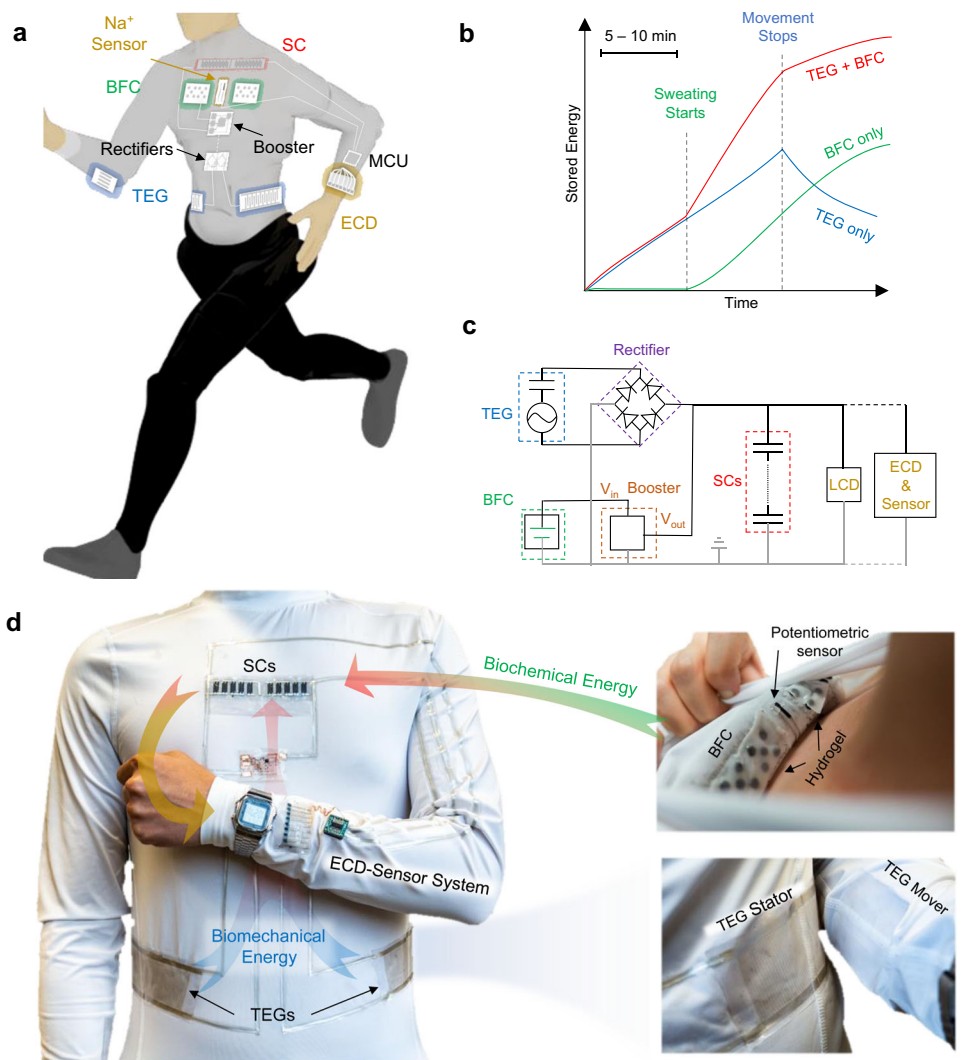

**Fig. 1 Design and concept of the multi-modular energy microgrid system. a** System diagram of the energy microgrid system, consisting of the TEG, BFC, SC modules and wearable applications. **b** Graphic illustration of the synergistic effect of integrating the complementary BFC and TEG energy harvesters. **c** System diagram of the integrated E-textile microgrid powering an LCD or an ECD-sensor system. **d** Photo images illustrating the arrangement of the individual modules of the wearable microgrid system on a shirt worn on-body, including the TEG modules on the side of the torso, the SC modules on the chest, the BFC modules and potentiometric sensor inside the shirt for direct sweat contact, and wearable electronics that are powered by the microgrid. All components were connected by printed stretchable silver traces insulated with SEBS.

realistic swinging speed of human arms from during walking (below 1 Hz) to running (1.5–3 Hz). As shown in Fig. 2c–d and Supplementary Fig. 6, the maximum peak voltage, independent from the frequency, was measured to be ca. 160 V. In contrast, the peak current grows nearly linearly with the frequency, from 45 µA at 0.833 Hz to 130 µA at 3 Hz, reflecting the linearly shortened time of transferring the same amount of charge between the mover and the stator[46]. To ensure that the TEG module is paired with commensurate storage rating, commercial electrolytic capacitors, with known capacitance ranging from 1 µF to 1 mF were used initially for the characterization, which were charged by a rectified TEG module in 60 s, 1.5 Hz sliding sessions while recording the capacitor voltage (Fig. 2e). To extrapolate the applicable performance data from the charging curves, the total harvested energy within the period and the harvested energy were calculated with the equation:

$$E = \frac{1}{2}CV^2 \qquad (1)$$

where $E$ is the stored energy in the capacitor in J, $C$ is a capacitance of the capacitor in F, and $V$ is the voltage of the charged capacitor in

$V$; and the average current was calculated with the equation:

$$I = C\frac{dV}{dt} \qquad (2)$$

where $I$ is the average current from the TEG module in A, $C$ is a capacitance of the capacitor in F, and $\frac{dV}{dt}$ is the rate of voltage change in V s$^{-1}$. Using the above equations, the total amount of stored energy and the average current was calculated and summarized in Fig. 2f. The average current remained mostly unchanged at 5.8 µA against different capacitors, while the energy stored maximized at 0.49 mJ near the load of 100 µF. The charging characteristics of the TEG modules at different frequencies were also tested and plotted in Fig. 2g–h. These data show a near-linear growth in stored energy and average current with the sliding frequency, which corroborated with the behavior observed in Fig. 2d. The mechanical durability of the TEG modules was also evaluated through repeated folding, crumpling and extended abrasion (Fig. 2i). The TEG modules were repeatedly folded with the radius below 1 mm for 100 cycles, and their open-circuit voltage (V$_{OC}$) during 1.5 Hz sliding motion was monitored, which

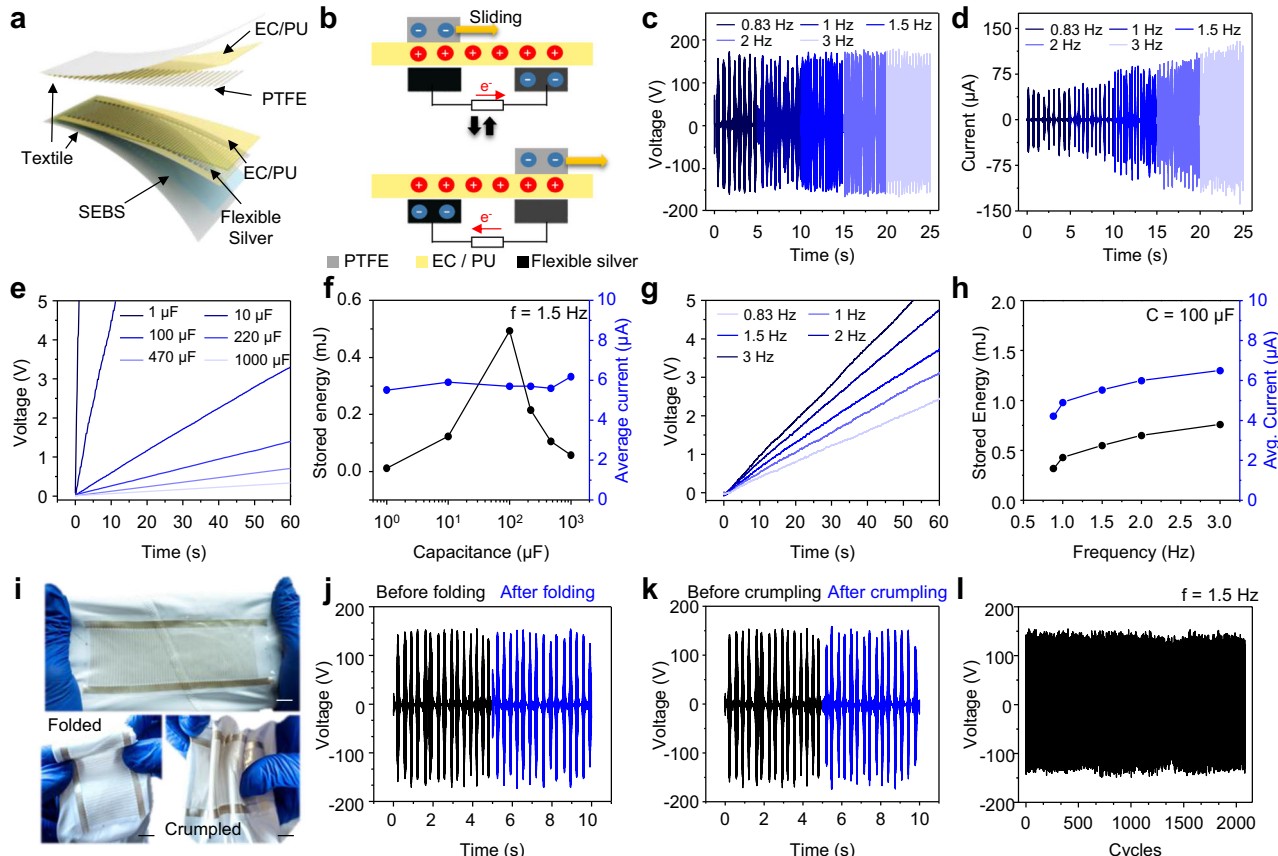

**Fig. 2 Characterization of the performance of the TEG module. a** Schematic image of the layer-by-layer composition of the textile-based TEG module composed of a PTFE-based mover (top) and an EC-PU-based stator (bottom). **b** The charge generation mechanism of the TEG module under in-plane friction between the mover and the stator. The unrectified output peak voltages (**c**) and peak currents (**d**) of the TEG at different sliding frequencies. The charging of commercial capacitors with difference capacitance at 1.5 Hz in a 60-s period (**e**) and the corresponding stored energies and equivalent average DC currents (**f**). The charging of 100 μF capacitor at different frequencies in a 60-s period (**g**) and the corresponding stored energies and equivalent average DC currents (**h**). **i** Images of folded and crumpled TEG stator. Scale bar, 1 cm. The unrectified output peak voltages before and after 100 cycles of folding (**j**) and crumpling (**k**). **l** The long-term electrical signal from the TEG module under continuous sliding motions with the frequency of 1.5 Hz.

has shown no discernible change (Fig. 2j). The TEG modules were also crumpled randomly and flattened repeatedly for 100 times, with their $V_{OC}$ when sliding monitored. The crumpling deformation had also shown no change in performance (Fig. 2k). The durability of the module against constant friction and washing was also tested, where the TEG module underwent constant sliding motion for over 2000 cycles and 20-min washing, where no visible drop in $V_{OC}$ was observed (Fig. 2l and Supplementary Fig. 21). The TEG modules could thus be considered a flexible and durable biomechanical energy harvest and were ready for integration. See Supplementary Note 1 and Supplementary Figs. 2–5 for the detailed device optimization, method of characterization, and performance calculation of the TEG.

In the wearable microgrid system, BFCs offer the feature of harvesting biochemical energy continuously from metabolites present in biofluids via electroenzymatic reactions. Due to the high lactate concentrations in human sweat, a variety of sweat-based BFCs have been developed as wearable energy harvesters[47–51]. The wearable BFC modules were also screen-printed using various ink composites onto textile substrates. Carbon nanotubes (CNT)-based pellets were attached to interdigitated "island-bridge" interconnections (Fig. 3a-b and Supplementary Fig. 8) composed of flexible carbon composite as currents collectors and flexible silver composite as conductive interconnections. As illustrated in Fig. 3c, such bioenergy harvesting relies

on the oxidation of lactate catalyzed by the lactate oxidase (LOx) immobilized on the bioanode, and on the oxygen reduction reaction facilitated by bilirubin oxidase (BOx) on the cathode. For efficient sweat bioenergy harvesting, the anode CNT pellets were preloaded with the 1,4-naphthoquinone (NQ) mediator while confining LOx with the glutaraldehyde (GA) cross-linker and chitosan; the CNT-based cathode pellets were decorated with protoporphyrin IX (PPIX) as the electron transfer promoter[52], while immobilizing the BOx with Nafion. The detailed fabrication and composition of the anode and cathode pellets are described in the "Methods" section and Supplementary Notes 2. All in-vitro tests for characterization of the BFC were carried out in 0.5 M phosphate buffer solution (PBS) with pH of 7.4. Traditionally, wearable BFC devices are characterized with LSV, with scan rate near 5 mV s$^{-1}$[11,16,49,52,53]. However, for high-surface-area electrodes like the CNT pellets such high-speed LSV will result in an unrealistically high power due to the capacitive currents (Supplementary Fig. 11). Therefore, the fabricated BFC modules were characterized using chronoamperometry (CA) at different potentials in the presence of 15 mM lactate to reflect the realistic power output of the BFC in extended discharge sessions. The testing of the BFC modules resulted in a maximum power of 21.5 μW per module when discharging the BFC at 0.5 V (Fig. 3d–e). The response of the BFC to different lactate concentrations has been evaluated using CA, as shown in Fig. 3f,

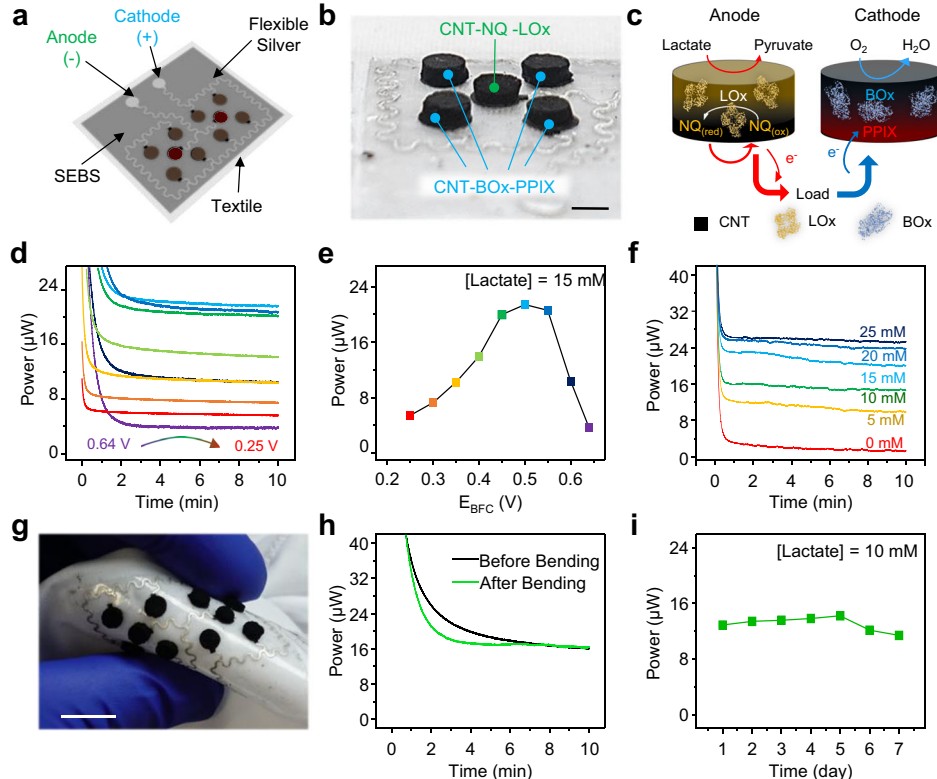

**Fig. 3 Characterization of the performance of the BFC module. a** Schematic image of the textile-based BFC module composed of eight BOx-based cathodes and two LOx-based anodes connected by printed flexible interconnections. **b** A zoomed-in photo image of the fabricated BFC module. Scale bar, 5 mm. **c** The charge generation mechanism of the BFC from lactate and oxygen in sweats. **d** The output power of the BFC module at different discharge potentials with 15 mM lactate concentration. **e** The equivalent polarization curve derived from (d). **f** The output power of the BFC module discharged at 0.5 V under various lactate concentrations. **g** Image of the BFC module under 180° outward bending. Scale bar, 1 cm. **h** The power of the BFC module before and after the 1000 bending cycles. **i** The output power of the BFC module with 10 mM lactate concentration everyday within a week.

Supplementary Figs. 12 and 13, where the power per module increases from 9.7 μW at 5 mM lactate to 25.3 μW at 25 mM lactate. The mechanical durability and the stability of the BFC were also tested using CA. The module was bent 1000 times to 180° inward then outward with the radius of 1.3 cm (Fig. 3g), with the current at 0.5 V before and after bending in 10 mM lactate environment measured. As shown in Fig. 3h, the power of the BFC did not show any noticeable change before and after the bending. Such resiliency is attributed primarily to the "island-bridge" structure where the non-flexible, functional electrode pellets as the islands are connected by flexible, conductive silver interconnections. The stability of the BFC was tested also throughout the week (as shown in Supplementary Fig. 14). The BFC under test was stored in refrigerator under 4 °C and was taken out for testing every 24 h in a 10 mM lactate environment. The results of the individual CA measurements are summarized in Fig. 3i. Additional details of the fabrication and characterization of the BFC module, including its washability and stability in simulated sweat conditions, can be found in Supplementary Note 2 and Supplementary Figs. 7-11, 15, and 21.

Flexible, printed SCs are selected for its ability to charge and discharge repeatedly and rapidly to deliver a flexible range of powers—a feature highly desirable for fast booting and pulsed high-power applications. A CNT and poly(3,4-ethylene dioxythiophene) polystyrene sulfonate (PEDOT:PSS) hybrid capacitor, offering screen-printability and high flexibility, has been adopted as symmetrical interdigitated electrodes that are connected by flexible silver current collectors (Fig. 4a and Supplementary Fig. 15)[54]. Each SC module consists of 5 SC units connected in series to reach the desired 5 V voltage range for directly powering

electronics, with each SC unit featuring four interdigitated CNT-PEDOT:PSS electrode segments covered by a solidified, transparent, flexible, sulfuric acid-crosslinked polyvinyl alcohol (PVA) electrolyte (Fig. 4b and Supplementary Fig. 16). The hybrid capacitor stores electric energy via both double-layer capacitance endowed by the high specific surface area CNTs and pseudocapacitance from the PEDOT:PSS, as illustrated in Fig. 4c. The areal capacitance of the printed SC material was characterized via both cyclic voltammetry (CV) at different scan rates and galvanostatic charge-discharge (GCD) with different currents. The GCD was conducted at charge/discharge between 0 V and 1 V with currents at 25, 50, 100, 250, and 500 μA (Fig. 4d). The capacitance at discharge is used to gauge the capacitance of the SC unit, which is calculated using the equation:

$$C = \frac{I\Delta t}{A(E_f - E_i)} \quad (3)$$

where $C$ is the areal capacitance in F cm$^{-2}$, $I$ is the current in A, $\Delta t$ is the time taken for the discharge, $A$ is the geometric area of electrodes, $E_f$ is the charged potential in V, and $E_i$ is the discharged potential in V. CV was carried out with the scan rates of 5, 10, 25, 50, 100 mV s$^{-1}$ between the window of 0 V and 1 V (Fig. 4e). The areal capacitance is calculated using the formula:

$$C = \frac{1}{2vA(E_f - E_i)} \int_{E_i}^{E_f} I \, dV \quad (4)$$

where $C$ is the areal capacitance in F cm$^{-2}$, $v$ is the scan rate in V s$^{-1}$, $A$ is the geometric area of electrodes in cm$^2$, $E_f$ is the higher vertex potential in V, $E_i$ is the lower vertex potential in V,

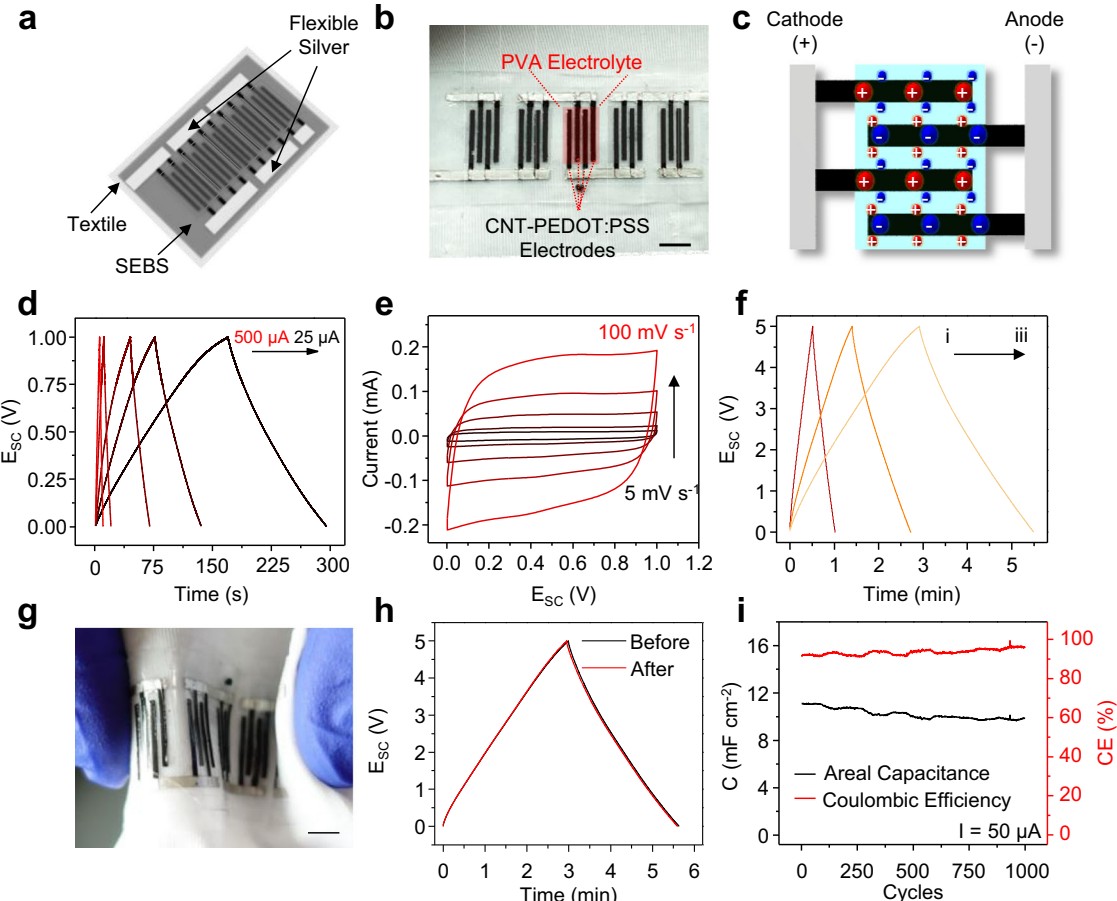

**Fig. 4 Characterization of the performance of the SC modules. a** Schematic image of the textile-based SC energy storage module consisting of current collectors, CNT-PEDOT:PSS-based electrodes and PVA-based solid gel electrolyte. Each module is composed of five SC units connected in series. **b** The photo image of the printed SC module. Scale bar, 5 mm. **c** The charge storage mechanism of the hybrid SC unit based on both the double-layer capacitance and pseudocapacitance. **d** GCD with currents of 25, 50, 100, 250, and 500 μA of a SC unit from 0 V to 1 V. **e** CV with scan rates of 5, 10, 25, 50, 100 mV s$^{-1}$ from 0 V and 1 V of a SC unit. **f** GCD of the energy storage modules with different configurations, with (i) two storage modules in series; (ii) one storage module; and (iii) two storage modules in parallel. **g** Images of the inward and outward bending of the SC module. Scale bar, 1 cm. **h** 5 μA GCD of the SC module before and after 1000 cycles of 180º bending deformation. **i** The areal capacitance and coulombic efficiency of a SC unit in 1000 charge-discharge cycles.

and the $I$ is the current in A. As agreed by both characterization methods, the capacitance of the printed SC was determined to be ca. 10 mF cm$^{-2}$ (Supplementary Fig. 17). Several SC modules can be connected in series or in parallel to adjust the overall capacitance to fit for harvesters and applications to obtain optimal charging speed and deliver sufficient energy (Fig. 4f). The stability of the module is analyzed by performing GCD cycles on the SC module. The mechanical stability of the module was evaluated by applying 1000 cycles of repeated inward-outward bending cycles at 180° with a bending radius of 0.5 cm (Fig. 4g). As shown in Fig. 4h, the charge-discharge behavior of the SC does not show any noticeable change before and after the 1000 cycles of bending deformation, suggesting the robust flexibility of the SC module for wearable applications. To prevent the leaching of the acidic electrolyte, the device is sealed by an additional layer of printed SEBS. The device can thus withstand extended wetting and washing without electrolyte leaching or observable degradation in its performance (Supplementary Fig. 21). The electrochemical stability of the SC module was studied by 1000 cycles of GCD between 0 V and 5 V with a current of 50 μA. As demonstrated in Fig. 4i, the areal capacitance of the SC shows a slight drop of areal capacitance after 1000 cycles to ca. 9 mF cm$^{-2}$, while the coulombic

efficiency gradually increases to 95%. The electrochemical performance of the SC is thus acceptable within the use-case of the microgrid, where roughly a few hundreds of charge-discharge cycles are expected. Detailed characterization and calculation of the SC performance are described in Supplementary Note 3 and Supplementary Figs. 18-19. Overall, each SC module was rated with the capacitance of ca. 150 μF.

**Synergistic bioenergy harvesting.** To demonstrate the synergistic effect of between TEG modules and BFC modules, a typical activity session has been simulated by applying constant 1 Hz–1.5 Hz sliding motions and spiking of 10–15 mM lactate fuel. Using this setting, three scenarios have been simulated: (i) movement starts followed by the start of perspiration; (ii) movements and perspiration taking place simultaneously; and (iii) movements stop but the perspiration continues. In the 4-min simulations, two TEG modules rectified by a bridge rectifier and one BFC module modulated by a DC voltage booster were used to charge one SC module (Supplementary Fig. 21). Each scenario was tested with only the TEG (blue curve), only the BFC (green curve), and both modules operating together (red curve). As shown in Fig. 5a, during the starting phase, the TEG module was able to charge the

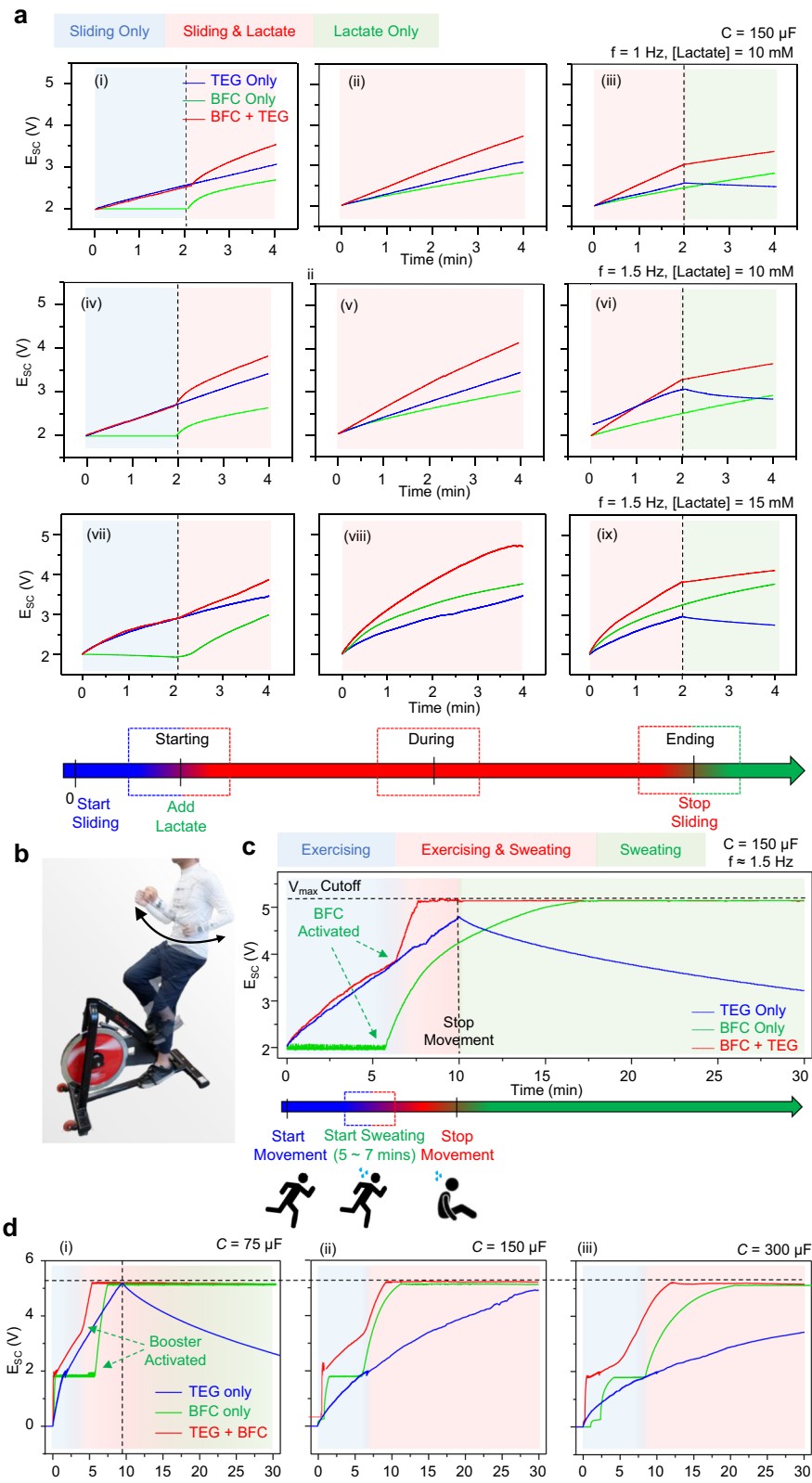

**Fig. 5 In-vitro and on-body charging performance of the wearable bioenergy microgrid system. a** In-vitro charging curves of the individual and integrated harvester with (i)-(iii) 1 Hz frequency and 10 mM lactate; (iv)-(vi) 1.5 Hz frequency and 10 mM lactate; and (vii)-(ix) 1.5 Hz and 15 mM lactate for the TEG modules and the BFC modules, which simulate three representative phases: starting ((i), (iv), and (vii)), exercising((ii), (v), and (viii)) and ending((iii), (vi), and (ix)) of an activity session. **b** Photo illustration of the on-body testing arrangement. **c** On-body charging of a 150 μF SC using the individual BFC or TEG module alone, and with both modules combined, in a 10-min exercise session followed by a 20-min resting. **d** On-body charging of capacitors from SCs with (i) 75 μF, (ii) 150 μF, and (iii) 300 μF from 0 V in a 30-min exercise session to represent the charging of the system including the cold-starting phase of the voltage-booster. During the 30-min period, movement stops once the SC if fully charged to 5.1 V.

SC module immediately after starting the movement, whereas the BFC module was not able to provide power due to the lack of lactate. When lactate was spiked, the BFC started to respond to the fuel addition to provide power to the SC. The additive effect of integrating two complementary energy harvesters can be observed throughout the rest of the simulation, where their charging speed surpasses the individually operating harvesters. Such additive effect continued throughout the scenario (ii) and the first half of the scenario (iii). As soon as the sliding motion stopped, the TEG module stopped its operation instantaneously. In contrast, the BFC module continued to operate due to the presence of the lactate fuel to charge the SC module without interruption. Beyond the additive effects of two energy harvesters, the advantage of integrating the complementary TEG and BFC modules was demonstrated: at the start of the movement, the fast-booting TEG module can compensate for the slow-booting of the BFC; in return, the transiently harvesting TEG module was compensated by the extended-operating BFC module after the movement stops, as illustrated in Fig. 1b. This synergistic behavior, not offered by any previous studies, is highly desirable for the reliable and sustainable operation of a self-powered wearable system. As expected, the charging speed of the 1 Hz sliding, 10 mM lactate simulation condition (Fig. 5a(i)-(iii)) was the slowest amongst all three, followed by the 1.5 Hz sliding, 10 mM lactate condition (Fig. 5a(iv)-(vi)), with the 1.5 Hz sliding, 15 mM lactate condition charging the capacitor fastest (Fig. 5a(vii)-(ix)). In all three situations, the synergistic additive effect was observed in all 3 phases of the condition, and the complementary fast-booting and extended-harvesting effects were observed for the starting and ending phases of all 3 conditions, correspondingly.

The synergistic effect of the integrated energy harvesting was further characterized with on-body tests to gauge the optimal capacitance that can mostly reflect such complementary fast-booting and extended-harvesting effects. Two TEGs, two BFCs and the corresponding SC modules were printed on the left and right sides of the waist, below the collar and in front of a shirt, respectively, as shown in Fig. 1c, and were connected via printed silver traces and enameled wires with proper insulations. A PVA-based PBS hydrogel was applied onto the BFC module for sweat capturing. A 10-min exercise session was carried out on a cycling machine, as illustrated in Fig. 5b, where the arm-swinging frequency was kept near 1.5 Hz, followed by 20 min of resting. Similar to the in-vitro simulation, the integrated on-body system was tested with only TEG, only BFC and with all harvesters operating together. Figure 5c demonstrates the on-body energy harvesting performance of the integrated system. For the system operating solely on TEG modules, the energy harvesting started immediately after the arm-swinging started and charged the SC module continuously throughout the 10-min exercise session. As soon as the movement stopped, the TEG stopped supplying power, and the SC module slowly self-discharged. For the system operated solely on the BFC module, the system suffered from slow booting with a 6-min delay. However, the BFC was able to supply power to the SC module quickly after sweating started, and fully charged the SC within 17 min Upon stopping the exercise, the BFC module was able to continuously supply power to the SC module over a 30-min period, reflecting the continuous presence of sweat. Lastly, the integrated system was able to compensate for both the slow booting and the transient harvesting, quickly fully charging the SC module in 7 min, maintaining maximum voltage over a 30-min period. In addition to the data presented in Fig. 5c, the on-body operation of the system was also tested with SC modules configured with the capacitance of 75 μF, 150 μF, and 300 μF (Fig. 5d). The charging is started at 0 V instead of 2 V to demonstrate the booting of the voltage booster in the start of the operation. The SC modules were fully discharged to 0V and their potential monitored during charging, and the exercise sessions were only stopped when the SC is fully charged to 5.1 V or the limit of 30 min was reached. As shown in the figure, the time taken to fully charge the SC module increased as the capacitance increased for both the individual harvesters and the integrated microgrid harvesters. The synergistic and complementary behavior can still be observed in the booting phase of the charging, where the integrated harvesting system can fully charge the SC module faster than the individual harvesters, starting from the 4 min for 75 μF capacitor, to the 8 min for 150 μF capacitor and 12 min for the 300 μF capacitor. These results have thus validated the in-vitro simulation, fully illustrating the complementary and synergistic behaviors of the wearable micro-grid system which offers fast-booting and extended-harvesting for energy harvesting in an activity session. Additional experimental details, data and discussion regarding the in-vitro and on-body tests can be found in Supplementary Note 4.

Two wearable applications were selected as examples of two operating modes for demonstrating the potential and advantages of the wearable microgrid system (Fig. 6a). The SC is an attractive energy storage module owing to its flexible discharge rates that allow powering of either low-power application continuously or of high-power application in a brief, pulsed fashion without damaging the module. For efficiently use the limited energy stored in the SC modules, the power consumption of the applications was characterized, and the SC modules were configured with minimum but sufficient capacitance for rapid booting while ensuring successful operation. Detailed fabrication and characterization procedures, additional data and further discussions can be found in Supplementary Notes 5 and Supplementary Figs. 22–31.

A textile-based sodium ion (Na$^+$) sensor integrated with a wearable, flexible ECD pixeled display was developed as an example for applications with higher power demand operating in pulsed mode. The developed potentiometric Na$^+$ sensor exhibited a near-Nernstian response to the concentration of the Na$^+$ target, with a potential change of 57.19 mV per decade of concentration change (Fig. 6d and Supplementary Fig. 23). The sensor output can be instantly read by the pre-programmed integrated circuit and reported by changing the color of individually controlled ECD pixels, as illustrated in Fig. 6c and Supplementary Movie 2. The operation of the ECD pixels follows the simple reversible redox reaction of the PEDOT:PSS,

$$PEDOT^+PSS^- + Na^+ + e^- \leftrightarrow PEDOT^0 + Na^+PSS^- \qquad (5)$$

that takes place when a voltage above +1 V is applied to the back-panel electrodes, and the reduction reaction take place on the front panel and turn its color from light blue to dark blue (Supplementary Movie 1). A low-power microcontroller unit (MCU) was selected for reading the input from the Na$^+$ sensor and controlling the on/off of the individual ECD pixels. The ECD is able to refresh rapidly and maintain the display without a continuous supply of power.

The set-up of the microcontroller is illustrated in Supplementary Fig. 28. The controller was pre-programmed to display potential up to 0.32 V, with each ECD pixel corresponding to one 0.04 V increment of the sensor output. The power consumption of the microcontroller connected to the sensor-ECD system was measured similarly using the potentiostat set at different voltage, as demonstrated in Supplementary Fig. 29. As shown, the power consumption of the microcontroller exceeded the wristwatch by 3 orders of magnitude, ranging from 4 mW at 2 V to 30 mW at 5 V. Yet, the microcontroller was able to boot quickly within the first 50 ms, take readings and apply signals to the ECD pixels within the first 200 ms, which allow this system to be powered transiently from the discharge of one charged capacitor. The energy consumed by one discharge session can be estimated by

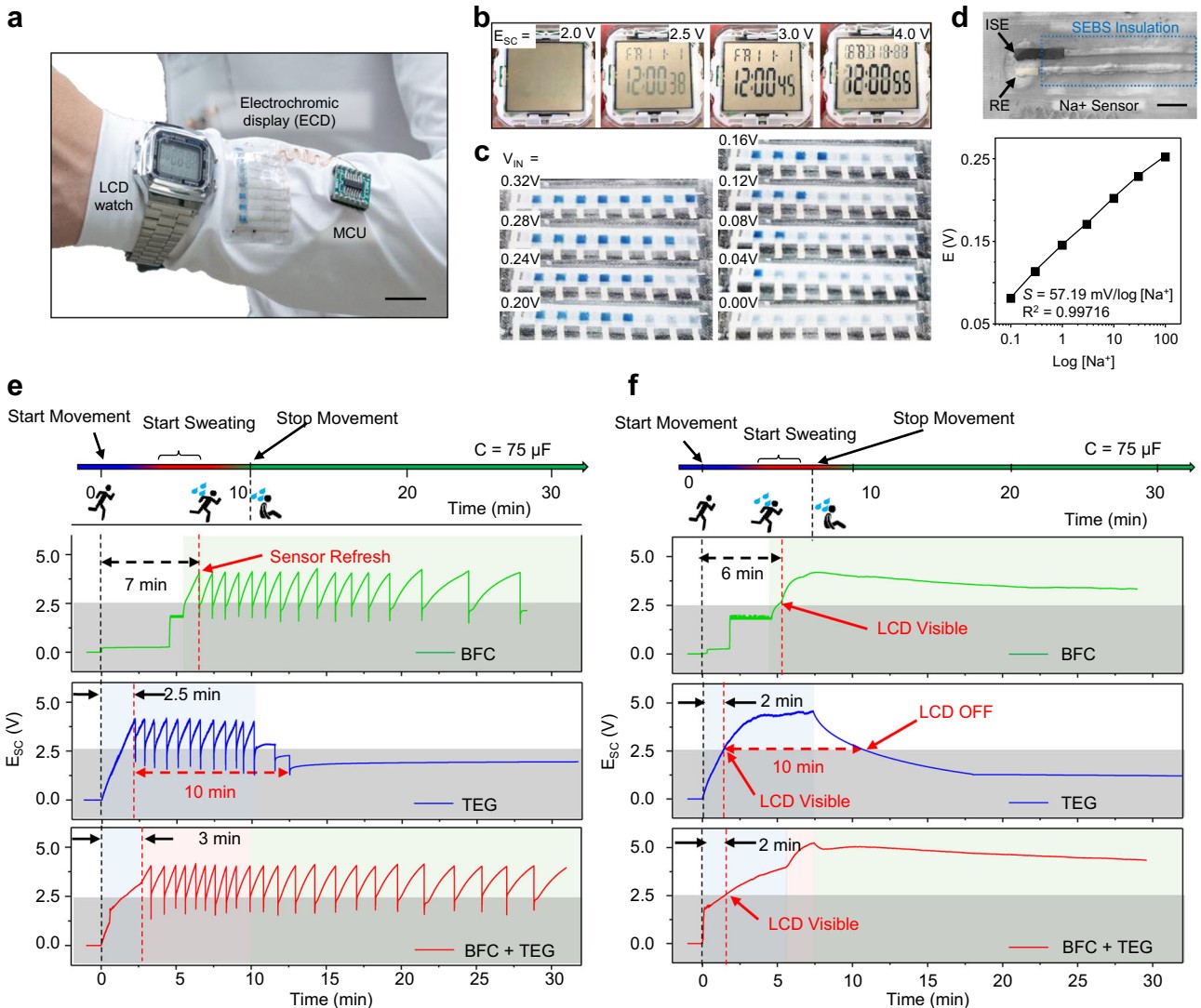

**Fig. 6 Wearable bioenergy microgrid system with different applications. a** Image of the LCD wristwatch and the ECD for the Na⁺ potentiometric sensor. Scale bar, 1 cm. **b** Images of the LCD screen with different input voltages. **c** The display of the ECD with different sensor voltage input. **d** The photo image of the all-printed flexible potentiometric sensor (top) and the calibration curve of the Na⁺ sensor (bottom). Scale bar, 2 mm. **e-f** Voltage vs. time curves of the wearable microgrid system powering the Na⁺ sensor-ECD system in pulsed mode during a 10-min running session followed by 20 min of rest, with only the BFC module operating, only the TEG module operating, and both harvesters operating together (**e**); and the LCD wristwatch continuously during a 7-min running session followed by 23 min of rest (**f**).

the equation:

$$E = P \times t \tag{6}$$

where $E$ is the energy in J, $P$ is the power in W, and $t$ is the time in s. To supply 200 ms of operation assuming the lowest power of ca. 4–5 mW, the energy required was thus estimated to be ca. 0.8–1 mJ. Capacitors ranging from 100 μF to 470 μF were able to supply sufficient energy to the microcontroller in one continuous discharge session with the larger capacitors had shown no substantial extended operation time (Supplementary Fig. 29b). The capacitors were discharged with 200 ms pulsed discharge, where more differences were more pronounced. The higher capacitance capacitors were able to sustain multiple refresh sessions, and the number of sessions decreases with the capacitance, as to 1 successful discharge session from the 100 μF with a slightly excess amount of energy, and 1 discharge from the 47 μF capacitor which did not last throughout the entire 200 ms session (Supplementary Fig. 29c). It is thus determined that the 75 μF SC modules should retain enough energy to sustain one

complete refresh session. The required state of charge of the SC was then characterized by charging the 75 μF SC module to different potentials, and use the SC to power one refresh session, while the color on the ECD recorded to determine the minimum state of charge required for the 75 μF SC module to induce color change with sufficient contrast (Supplementary Fig. 30). It can be observed that the contrast of the on and off pixels gradually decreased with the initial state of charge of the SC modules, with the difference barely recognizable below 3.5 V. The minimum potential of 4 V before initiating a refresh session was thus decided. To confirm with the calculation above on the energy required for one operation session, the energy stored in the capacitor was calculated using equation:

$$E_{SC} = \frac{1}{2} C \left( V_i^2 - V_f^2 \right) \tag{7}$$

where $E_{SC}$ is the energy discharged from the SC in J, $C$ is the capacitance of the SC in F, $V_i^2$ is the potential of the SC before the discharge in V, and $V_f^2$ is the potential of the SC after the

discharge in V. For a 75 µF SC to discharge from 4 V to 1.25 V, the energy released from the discharge is thus calculated to be 0.88 mJ, which agreed with the previous calculation based on Eq. (6). The 75 µF SC modules with a threshold voltage of 4 V was thus selected to allow fast booting while providing sufficient energy for a successful sense-refresh session.

The integrated wearable sensor-ECD application was integrated into the microgrid and tested on the body in a 10-min exercise session. As shown in Fig. 6e, the SC module was charged up to 4 V and discharged to refresh the sensor-ECD system. The fully integrated microgrid system was able to boot quickly within 3 min and intermittently refresh the results during the 30-min operation. In comparison, the BFC module suffered from slow booting due to delayed perspiration while the TEG modules were not able to sustain the operation after stopping the movement. It is worth noting that the faster boosting speed for the TEG-only scenario is due to the lack of need to supply a small amount of energy to boost the voltage booster. In both applications with different modes of operation, the wearable microgrid system—with its complementary and synergistic BFC-TEG harvesting and commensurate SC pairing—was able to deliver both fast-booting and extended-harvesting to ensure the autonomous and sustainable operation of the wearable platforms.

A low-power wristwatch with a LCD was chosen as a representative application that is continuously powered by the wearable microgrid system. The voltage of 2.5 V is determined as the turn-on threshold voltage, corresponding to the minimum voltage for the LCD to display with sufficient contrast (Fig. 6b). The power consumption of the watch was rated below 10 µW which has low requirement in storage unit (Supplementary Fig. 27). Two SC modules connected in-series with the capacity of 75 µF were selected to offer fast booting while maintaining extended operation when the microgrid harvests from human activities. Short, 8-min exercise sessions were carried out, with the potential of the SC recorded continuously. As illustrated in Fig. 6f, the simultaneous BFC-TEG energy harvesting was able to quickly boot the wristwatch within 2 min and maintain its continuous operation for over 30 min In contrast, the BFC-only system suffered from slow booting while the TEG-only system can only maintain the operation for a short time period before the SC module voltage dropped to below 2.5 V.

## Discussion

In summary, we have demonstrated the concept of wearable bioenergy microgrid via a textile-based multi-module system for sequentially harvesting biomechanical and biochemical energy via the TEG and BFC modules. The microgrid can store and regulate the harvested energy via efficiently paired SC modules to efficiently power wearable applications such as an LCD wristwatch and a sensor-ECD system. Implementing the "compatible, complementary, and commensurate" design principles, all modules were carefully characterized and efficiently integrated to rapidly, autonomously and sustainably deliver power using the limited amount of harvested energy. All printed modules, printed with polymer-based composite inks, are flexible, durable, and ready for seamless integration on textile platforms. Compared to previous integration of wearable energy harvesting and storage devices, this work focuses on the complementary relationship between two bioenergy harvesters that perform synergistically to scavenge energy from human motion, and their pairing with storage modules with commensurate capacity for maximized efficiency and performance. With the optimized pairing of components, such wearable microgrid can quickly boot selected applications while sustaining their operation substantially longer than the exercise sessions, hence ensuring the reliability and practicality of

the wearable system. To further improve such E-textile microgrid system, more modular design concepts, such as fabricating patch-like detachable and swappable harvesters, storage devices, and sensors, can be adapted to extend the usage and applicable scenario of the E-textile system. While serving here as an example, such microgrid strategy inspires future miniaturized integrated systems that harvest thermal, chemical or mechanical energies to select complementary, synergistic, commensurate and compatible components for their designated use case. The configuration of the harvesters connection can also be further explored: instead of parallelly operating harvesters, serial connections between harvesters of the same type or different types can be considered to increase the output voltage and minimize or eliminate the need for voltage regulation. Future work in developing energy harvesting devices that rely on the harvesting from passive activities will broaden the applicable scenarios and enhance the practicality of the microgrid system. Expanding the microgrid concept from on-body to in-body applications, and utilizing the abundant amount of biofuel and biomechanical movements within the body can also allow integrated harvesting system for self-powered implantable or ingestible sensing that are no longer limited by external environment or the need for exercise. Upon improving the power density of various harvesters, high capacity energy storage options with commensurate rating can also be considered for powering of a wider range of electronics for advanced functionalities. The concept of wearable microgrids and their design principles are thus expected to promote system-level considerations for integrating various new-form-factor modules towards truly self-powered, autonomous, and sustainable systems.

## Methods

**Chemicals and reagents**. 1,4-NQ, bovine serum albumin (BSA), PPIX (≥95%), glutaraldehyde, L(+)-lactic acid, acetic acid, sulfuric acid, chitosan (from shrimp shells, >200 cP, 1 wt. % in 1% acetic acid), silver flakes (10 µm), PTFE powder (powder, free-flowing, 1 µm particle size), toluene, methanol, ethanol, PVA (MW = 89,000 – 98,000, 99+ % Hydrolyzed), polyvinyl butyral (PVB) (powder, >100 cps), polyvinyl chloride (PVC), 4-methyl-2-pentanone (MIBK), 1,1,1,2,3,4,4,5,5,5-decafluoropentane (DFP), Poly(sodium 4-styrene sulfonate) (MW ~ 70,000), sodium tetrakis[3,5-bis(trifluoromethyl)phenyl]borate (Na-TFPB), D-sorbitol, dioctyl sebacate (DOS) and glycerol were purchased from Sigma-Aldrich (St. Louis, MO, USA). L-lactate oxidase (LOx) (80 U mg$^{-1}$) was purchased from Toyobo (Japan). BOx (>1.2 U mg$^{-1}$) was obtained from Amano Enzyme. Carboxyl-functionalized multi-walled carbon nanotubes (MWCNT-COOH, Ø = 10-20 nm, 10-30 µm length, >95% purity) were purchased from Cheap Tubes Inc. Water-based polyurethane resin Dispercoll U-42 was purchased from Covestro (Germany). Tetrahydrofuran (THF) was purchased from EMD Millipore. Ethylcellulose (EC) (Standard 4), polyurethane (PU) (Tecoflex SG-80A) were obtained from Lubrizol Life Sciences. SEBS (G1654) was obtained from Kraton (TX, USA). Fluorinated binder (T-70, a terpolymer of vinylidene fluoride, tetrafluoroethylene and hexafluoropropylene) was obtained from Frechem (Jiangsu, China). Super-P carbon black was purchased from MTI Corporation (Richmond, CA, USA). Graphite powder was purchased from Acros Organics (USA). Stretchable textile Lycra Shiny Milliskin Nylon Spandex Fabric was purchased from Spandex World. Inc (USA). The Capstone$^{TM}$ fluorosurfactant FS-65 is purchased from DuPont (USA). The graphite carbon paste was purchased from Ercon Inc. (Wareham, MA, USA). The screen printable PEDOT:PSS ink (C2100629D1) was purchased from Sun Chemical Ltd. (Bath, UK). All metal stencils were designed via Autodesk AutoCAD (CA, USA) and are ordered from Metal Etch Services (San Marcos, CA, USA). The voltage booster (Texas Instruments bq25505) and the microcontroller (Atmel AtTiny441) were purchased from Digi-Key Electronics (MN, USA). The wristwatch (Casio, A178WA-1A) was purchased from Amazon.com (WA, USA).

**Formulation of flexible polymer composite Ink**. The SEBS resin was prepared by dissolving 4 g of SEBS polymer in 10 mL toluene. The silver ink was made through a method similar to our previous protocol[48,55]. Briefly, flexible silver composite ink was formulated by mixing silver flakes with the SEBS resin (weight ratio = 2: 1) by a dual asymmetric centrifugal mixer (Flacktek Speedmixer, DAC 150.1 KV-K) for 5 min with a speed of 1800 rotations per minute (RPM). The flexible carbon composite ink is formulated by mixing super-P powder, graphite powder, SEBS resin, and toluene (weight ratio = 1: 6: 8.4: 2.1) in the mixer for 5 min at 2150 RPM. The PTFE ink was formulated by mixing the PTFE powder, T-70, DFP and MIBK (weight ratio = 9: 6: 2: 5) in the mixer for 10 min at 1800 RPM. The EC-PU ink was formulated by mixing EC powder, PU and THF (weight ratio = 1: 1: 14) in

the mixer for 5 min at 2000 RPM. The CNT-PEDOT:PSS ink for the SC modules was formulated by mixing the MWCNT-COOH, PEDOT:PSS paste and fluor-osurfactant (weight ratio = 5: 95: 0.5) in the mixer for 5 min at 2500 RPM. The PVA-acid electrolyte gel was formulated following our previous work by dissolving 1 g of PVA into 10 mL of water, adding 10 g of 1.8 M sulfuric acid and heating with rigorous stirring with a magnetic stir bar on a hot plate at 100 °C until 50 % of the water has evaporated. The ink formulations for the electrodes and electrolyte used in the ECD are discussed in detail in Supplementary Note 5.2.

**Fabrication of flexible TEG modules**. The whole layers of flexible TEG modules were mainly made into a 100 μm-thick metal stencil for printing. The stator of the TEG module is fabricated by firstly casting a layer of SEBS lining with the SEBS resin onto the textile substrate using an adjustable doctor blade with a thickness of 200 μm and cured in an oven at 65 °C for 10 min Then, the interdigitated silver current collector layer was printed using the flexible silver composite ink and the designed metal stencil and cured at 60 °C for 20 min After that, a positively charged layer based on EC-PU ink was printed using the doctor blade with a thickness of 50 μm and cured at 60 °C for 10 min For mover of the TEG, the EC-PU layer was first printed on the surface of the textile using a doctor blade with 100 μm thick and cured at 60 °C for 10 min as lining. And then, interdigitated, negatively charged layer was printed using the PTFE ink with the metal stencil and cured at 70 °C for 20 min

**Fabrication and assembly of BFC modules**. The CNT pellets for the electrode were prepared by formulating a paste that was molded with a thick PTFE stencil. The formulation is modified based on an early publication[11]. The anode paste was formulated by blending 7 mg MWCNT-COOH, 2 mg NQ, 11 μL GA (1% in ethanol) and 56 mg chitosan solution (3 wt% in 0.1 M acetic acid) with mortar and pestle. The cathode past was formulated by blending 8 mg MWCNT-COOH, 11 μL glutaraldehyde (1 % in water) and 56 mg chitosan solution (3 wt% in 0.1 M acetic acid) with mortar and pestle. The paste was then molded using a PTFE stencil and dried in the oven at 80 °C for 20 min to form pellets.

The design of the BFC current collector array was made into a 100 μm-thick metal stencil for printing. To fabricate the current collector, a layer of SEBS resin was firstly cast onto the textile substrate using an adjustable doctor blade with a thickness of 200 μm and cured in an oven at 80 °C for 10 min The interconnects are then printed using the flexible silver composite ink and cured at 80 °C for 10 min The flexible carbon composite ink was then printed as the cathode and anode current collectors and cured at 80 °C for 10 min The anode and cathode pellets were bonded on the carbon current collectors by printing a layer of commercial carbon paste use the metal stencil, mounting the anode and cathode pellets accordingly, and curing the paste at 60 °C for 20 min To solidify the adhesion between the pellets and current collectors, the water-based PU was coated on the conjunctions and dried at 60 °C for 20 min

The anode was functionalized by drop-casting 5 μL of LOx (20 mg mL$^{-1}$ in 10 mg mL$^{-1}$ BSA), 5 μL of GA solution (1 % in ethanol) and 5 μL of chitosan (1 wt% in 0.1 M acetic acid) on each pellet. The functionalization of the cathode is similar to that of the anode by drop-casting 5 μL of PPIX (40 mM in 9:1 vol/vol ethanol/acetone), 5 μL BOx (40 mg mL$^{-1}$ in 10 mg mL$^{-1}$ BSA) and 2.5 μL Nafion (1 wt% in ethanol) on each pellet. Each step of drop-casting was separated by 5 min to allow the solvents to evaporate, with chitosan and Nafion used for confining the LOx and BOx, respectively. Lastly, the modified BFC module was left in the refrigerator at 4 °C overnight before use. More details on the fabrication of the BFC modules are discussed in detail in Supplementary Note 2.

The fabrication of the PVA hydrogel for the BFC on-body tests were adapted from a previous work[49]. Shortly, a 20 wt% PVA solution was prepared by dissolving the PVA in water in an 80 °C water bath, and a 10 wt% KOH aqueous solution was prepared and added to the PVA solution at 1:1 ratio by weight to form a hydrogel precursor. The precursor was left in desiccator under vacuum for 1 day until crosslinked, and then taken out to then be washed in DI water to remove excess KOH. The washed hydrogel was soaked in 0.5 M (pH 7.4) PBS for later use. The formed hydrogel was measured to be ~0.5 mm thick and was cut into 2 cm by 4 cm pieces to be used for each BFC module.

**Fabrication of flexible SC modules**. The design of the SC modules was made into a 150 μm-thick metal stencil for printing. First, a layer of SEBS resin was cast onto the textile substrate using an adjustable doctor blade with a thickness of 200 μm and cured in an oven at 80 °C for 10 min The CNT-PEDOT:PSS ink was printed on the stretchable textile as the electrodes and is dried at 60 °C for 10 min and 80 °C for 10 min The flexible silver composite ink was then printed onto the electrodes as current collectors to connect electrodes in series and cured at 80 °C for 10 min The PVA-acid gel was printed onto the electrodes as the electrolyte and left to dry in a well-ventilated place under room temperature overnight. A layer of SEBS resin, which forms a dense, hydrophobic, water-proof layer upon drying, was printed onto the SC after drying to seal and protect the solidified gel electrolyte from leaching.

**Fabrication of flexible Na$^+$ sensors and electrochromic display**. The details of the fabrication and characterization of the flexible sodium sensor and the ECD are discussed in Supplementary Notes 5.1-5.2 and Supplementary Fig. 22-26.

**Design and fabrication of the energy regulation and sensing-display circuits**. The ultra-low-power boost charger circuit was developed based on the bq25505 chip (Texas Instruments). The circuit was configured to discharge the connected BFC at 0.5 V and charge the connected SC modules up to 5.1 V. The sensing-display circuit was developed based on the energy-efficient AtTiny441 micro-controller (Microchip Technology). The on-chip 10-bit A/D converter provides the capability of potential reading. Ten digital I/Os was configured for the ECD control. The design schematics and circuit connections were discussed in Supplementary Notes 4.3 and 5.4 and illustrated in Supplementary Fig. 20, 21, 28 and 31.

**Characterization of the integrated devices**. An MSO-X-3014A oscilloscope (Agilent Technologies) and the designed OP-Amp circuit operated by a power supply (−10 V ~ 10 V) were used to measure the output voltage and currents of the TEG modules under various frequency from 0.88 ~ 3 Hz. Commercial electrolytic capacitors (1 μF ~ 1 mF) were used to characterize the average power, current of TEG. An Autolab PGSTAT204 potentiostat/galvanostat and an Interface 1010E potentiostat (Gamry Instruments, PA, USA) were used to analyze the electro-chemical performance of the BFC, SC, sodium sensor and the ECD modules.

The on-body test was performed by printing/glueing the individual modules onto a long-sleeve polyester shirt (Starter Dri-Star) and connected by printed, stretchable silver traces and enameled copper wire (36 AWG). The printed traces and the contact points of the enameled wires were further insulated with SEBS to avoid short-circuiting from sweat. The human subject volunteers were informed of the on-body experiment details and asked to sign the consent. The whole process has strictly followed the guidelines of Institutional Review Boards (IRB) and approved by Human Research Protections Program at University of California, San Diego. Further On-body testing setup information can be found in Supplementary Note 4.3 and 5.4.

## Data availability
The data that support the plots within this article and other findings of this study are available from the corresponding author upon reasonable request.

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

## Acknowledgements

This research was supported by the UCSD Center of Wearable Sensors (CWS) and the National Research Foundation of Korea (NRF-2018R1A6A3A03011252 and NRF-2019R1A6A3A12033345). We would like to thank Kraton Corporation and Amano Enzyme for providing all the SEBS samples and the BOx enzyme sample, respectively.

## Author contributions

L.Y., K.N.K., and J.L. equally contributed to this work. L.Y., K.N.K., J.L., and J.W., conceived the idea, designed the experiments, guided the projects, and wrote the manuscript. L.Y., K.N.K, and J.L. conducted experiments. J.M. and J.-M.M. fabricated samples, participated in figure design and revised the manuscript. J.Y. fabricated samples. F.T. and J.Y. fabricated samples. M.L. and Z.L. designed and fabricated circuits for electronics. S.X. provided suggestions to experiment designs, figure designs, and revised the manuscript.

## Competing interests

The authors declare no competing interests.
