## [Peer Review File · Nature Communications]

Reviewer #1 (Remarks to the Author):

Thank you for the opportunity to review this groundbreaking article by Prof Jo Wang et al on a wearable multi-modular e-textile system that exploits triboelectric, biofuel cell and supercapacitor devices of powering multiple electronic devices and including real experiences on the human body. The publication very effectively summarizes a very large body of work that is sufficiently detailed in terms of experimental detail.

The development of energy harvesters e.g. for wearables is crucial for next generation sustainable technology. However, the various energy harvesting technologies available all have their various limitations. As such, although complex, the development of multi-modal energy sources is a great solution that will help with a shift away from, for example, toxic battery chemistries.

The specific combination of a triboelectric generator with an enzymatic biofuel cell is original, and of course, as a three power source system with the triboelectric generator, biofuel cell and supercapacitor.

The experiments were carefully performed demonstrating clearly the concepts proposed by the authors. Taking into account the originality of the concept, I recommend its publication with minor revision.

Please find below critical comments

1. Abstract. "relies on solely on human movements". The sentence is not entirely clear as the TEG relies on physical movement and the fuel cell and capacitor on chemical/biological/charge movement. Something like "relies solely on human activity" seems better with physical and biochemical activities.
2. Abstract. Many positive points mentioned like "fast" and "extended" and "efficiently" and "pulsed sessions" but what are the key performance characteristics and/or conditions? How fast, how extended, etc.
3. Intro. Biofuel cell may be misleading as typically refers to biological materials transformed into fuels to be consumed. In this work, natural fuel/oxidant is used from biological sources.
4. Intro. Parallel connection of harvesters is discussed but what about serial connection, this should be considered
5. The definition of a microgrid could perhaps be more clear for general readership
6. Intro. A 2020 glucose fuel cell-TEG device has recently been reported, this article should be cited if not already done so.
7. The authors refer to enzymatic reactions but it is technically correct to refer to "electroenzymatic" reactions.
8. The authors refer to the terminology "complimentary, commensurate, compatible". Is this a new terminology or previously reported. If previously reported, citation is required.
9. Fig 1b is missing values of time on the x axis, at least approximate ones.
10. What is the SEBS abbreviation, should be discussed near to first mention.
11. Fig 2b too small/unclear
12. What is the mechanical performance of the devices during folding? The authors report before and after folding and crumpling cycles but not during? Why were such experiments not performed as the more realistic way to test in vivo operational performance?
13. 0.833 Hz to 3 Hz is the swinging speed of human arms, ref?
14. Fig 3b. CNT-LOx should be CNT-NQ-LOx?
15. BOD would be better as BOx (LOx is used not LOD in the article) for consistency
16. Fig 3. Difficult to read figures e.g. d and f due to similar green line curves. Labels or colours need to be improved for clarity.

17. The testing solution, pH and temperature values should be included in Figures . I understand that it may have always been 0.1M phosphate buffer pH 7.4. for in vitro experiments (capacitance, voltammetry)? If so, this can be repeated a few times or PB or PBS used periodically in the text/figure captions. Page 8. More scientific details for the CNT based biopellet preparation/composition in the text would be appreciated/valuable, as an integral part of the biofuel cell.
18. Page 9. The BFC was stored in in the presence or absence of lactate?
19. Page 11. Which area, geometric?
20. How much volume is exactly from the sweat using the PVA-based PBS hydrogel and what is the thickness? Important and interesting details to include in the main publication.
21. Device powering was continuously demonstrated for 30 min. Is it possible to power for longer or using the same device on different days?
22. Methods. What are the molecular weights of all polymers used, and enzyme activities for BOx and LOx?
23. Methods. For BFC module, how long for the GA crosslinking reaction before drying the sample? Why this time? This detail seems to be missing.
24. Methods. When dropcasting LOx, why was BSA and why chitosan at the anode and nafion at the cathode?
25. SI -the authors used 1 M KCl for half cell characterization of bioelectrodes? Is this not a harsh environment for the enzymes leading to destabilization and deactivation?
26. Figure S10. It is not obvious that there is catalytic oxygen reduction?

Reviewer #2 (Remarks to the Author):

In this manuscript entitled authors have discussed the integration of textile based microgrid systems for energy efficient, sustainable, and autonomous wearables. The fabrication of TEG, BFC and SC are discussed in detail and their integration for self-powered applications are also demonstrated by powering wearable sensors. Overall, I find this work as interesting with sufficient details related to scientific and technological aspects. As far the integration of self-sustained e-textile based bioenergy system is concerned, the work has merits and can be considered for publication after minor revision by addressing the following comments.

Some findings are given below:

1. In figure 2 caption and following discussion authors referring capacitor instead of supercapacitor. To avoid confusion please correct it to SC.
2. In BFCs characterization authors varied the lactate concentration and measured the performance of the device by analysing the power output. However, in real human sweat apart from lactate other ions like sodium, potassium, urea, amino acids etc are present. Do these ions have any influence on the BFC performance? Did author check the performance of BFC device with different conc. of these ions (with sweat equivalent solution)?
3. In SC fabrication authors used sulfuric acid as the gel electrolyte. However, for wearable application is concerned, the strong acid-based electrolyte may cause some user discomfort. Please comment on it?
4. In Figure 5 (a) iii and vi, the TEG device shows a constant voltage output (instead of decreasing) in lactate only region (without sliding). what is the reason for maintaining this constant voltage for TEG device even without any applied mechanical (sliding) force?

5. Did authors check the performance reliability of the whole micro-grid integrated cloth (having TEG, BFC and SC) under different washing conditions. Since coming to the real time application the stability of the device against washing cycles may be a matter.

Reviewer #3 (Remarks to the Author):

In this manuscript, the authors attempted to develop a self-powered, multi-functional micro-grid system. Triboelectric generators and biofuel cells were utilized as the energy harvesting module, and the supercapacitors were selected as the energy storage module for the construction of the wearable E-textile. They have also conducted many experiments to prove the complementary, commensurate and compatible features of each functional module. The manuscript is clearly structured and well written. Below are some suggestions for the authors to further increase the manuscript quality.

1. The idea is interesting, and I would like to suggest the authors to more highlight the advantage of this design compared with the previous reports. Some related information may be found here, *Angew. Chem.* 10.1002/ANIE.201207023; *Adv. Mater.* 10.1002/ADMA.201302951.

2. Combining the TEG and BFC modules to efficiently scavenge energy from human activities is nice. While would these two modules influence each other in realistic scenarios? For instance, the BFC module worked with the existence of a lot of sweat in the textile, would this affect the triboelectric performance of the TEG?

3. Most of the data for the BFC were recorded in the lactate solutions. While in realistic application, many factors including the temperature, pH and surface contamination, could significantly affect the BFC behavior, which may be helpful to be considered.

4. The authors claimed that the capacitance of each SC was $\sim 150 \mu\text{F}$, which may be limited to power various electronics. Will it be possible to integrate flexible batteries with high capacities into this E-textile system?

5. The BFC and TEG modules can only collect ambient mechanical energy at sports mode while most people need to stay quietly to work. I wonder about the possibility of an energy harvesting system that requires lower dependency to environment and usage scenario. Some comments may be helpful to the readers.

Detailed list of changes made and Responses:

The paper was carefully revised to address both the reviewers' and editor's comments; the specific changes and comments can be found in the listing below. We believe that these changes have further improved this paper.

Response to Reviewer #1

General Comment:

Thank you for the opportunity to review this groundbreaking article by Prof Jo Wang et al on a wearable multi-modular e-textile system that exploits triboelectric, biofuel cell, and supercapacitor devices of powering multiple electronic devices and including real experiences on the human body. The publication very effectively summarizes a very large body of work that is sufficiently detailed in terms of experimental detail.

The development of energy harvesters e.g. for wearables is crucial for next generation sustainable technology. However, the various energy harvesting technologies available all have their various limitations. As such, although complex, the development of multi-modal energy sources is a great solution that will help with a shift away from, for example, toxic battery chemistries.

The specific combination of a triboelectric generator with an enzymatic biofuel cell is original, and of course, as a three power source system with the triboelectric generator, biofuel cell and supercapacitor.

The experiments were carefully performed demonstrating clearly the concepts proposed by the authors. Taking into account the originality of the concept, I recommend its publication with minor revision.

Please find below critical comments

Response:

We would like to thank the reviewer for recognizing the quality and novelty of our work.

Comment 1

Abstract. "relies on solely on human movements". The sentence is not entirely clear as the TEG relies on physical movement and the fuel cell and capacitor on chemical/biological/charge movement. Something like "relies solely on human activity" seems better with physical and biochemical activities.

Response:

We thank the reviewer for this helpful comment. Wearable BFC most commonly relies also on physical activity. We revised the Abstract to ‘human activity’.

The manuscript has thus been updated on page 1 to address this suggestion.

Comment 2

Abstract. Many positive points mentioned like “fast” and “extended” and “efficiently” and “pulsed sessions” but what are the key performance characteristics and/or conditions? How fast, how extended, etc.

Response:

We would like to thank the reviewer’s critical comment that helps improve the quality of the abstract.

The abstract has been modified (page 1) to include quantitative descriptions of key improvements.

Comment 3

Intro. Biofuel cell may be misleading as typically refers to biological materials transformed into fuels to be consumed. In this work, natural fuel/oxidant is used from biological sources.

Response:

We thank the reviewer for raising this question about the usage of the word “biofuel cells”. Different from the concept of biofuels, which usually discusses the synthesis or extraction of fuels from biological sources, the term “biofuel cell” has been specifically used to describe fuel cells that use biomaterials (enzymes, bacterial, etc.) as catalysts. This concept has been around for several decades since the ’60s and was generally accepted as a standard term in the field of electrochemistry.[1-3] We recognize that this term may cause confusion for the broad spectrum of Nature Communication’s audience.

Accordingly, the “biofuel cells” has been changed to “enzymatic or microbial biofuel cells” on page 2 to clarify the type of device discussed in this work.

[1] Science, 1962, 137, 615-616.

[2] Electroanalysis, 2016, 28, 1188-1200.

[3] Adv. Funct. Mater. 2020, 30, 1906243

Comment 4

Intro. Parallel connection of harvesters is discussed but what about serial connection, this should be considered

Response:

We would like to thank the reviewer for this insightful comment. In the introduction on page 2, the term parallel was used figuratively to describe that the harvesters work independently without interfering with each other. In most of the work integrating different harvesters, due to their difference in voltage, generation mechanism (AC or DC), input type, etc., usually, they require different regulation circuits and are generally not connected directly in parallel. There have been studies that connect harvesters of the same type (e.g. solar cells, BFCs, thermoelectric generators) in series which can increase the voltage of the harvesters. However, in general, harvesters with different generation mechanisms have their own limitations, to connect harvesters in series means that the system is limited by the combination of all disadvantages and will not work unless all harvesters are operational. In this situation, the serial connection of harvesters is not desirable.

To address this comment, we have included additional discussion about the possibility of a serial connection of harvesters in the Discussion section of the manuscript on page 22.

Comment 5

The definition of a microgrid could perhaps be more clear for general readership

Response:

We would like to thank the reviewer for this helpful suggestion.

We have included additional discussion about the concept of microgrids for the general readership to page 2 of the manuscript.

Comment 6

Intro. A 2020 glucose fuel cell-TEG device has recently been reported, this article should be cited if not already done so.

Response:

We thank the reviewer for this kind suggestion. We are aware of a work of integrated glucose BFC and TEG device for proposed subdermal implantation has been demonstrated to operate in in-vitro settings.[1]

This work was already cited as ref 39 (now ref 41).

[1] Nano-Micro Lett. 2020, 12, 1–12.

Comment 7

The authors refer to enzymatic reactions but it is technically correct to refer to “electroenzymatic” reactions.

Response:

We thank the reviewer for this helpful suggestion. The manuscript has been updated on pages 3 and 9 to include this accurate description: “BFCs harvest biochemical energy from the electroenzymatic reactions” and “BFCs offer the feature of harvesting biochemical energy continuously from metabolites present in biofluids via electroenzymatic reactions”.

Comment 8

The authors refer to the terminology “complimentary, commensurate, compatible”. Is this a new terminology or previously reported. If previously reported, citation is required.

Response:

We thank the reviewer for this question. The design concept of complementary behavior, commensurate energy rating, and compatible form factor is firstly proposed in this work; it has not been previously reported based on our knowledge.

Comment 9

Fig 1b is missing values of time on the x axis, at least approximate ones.

Response:

We thank the reviewer for this useful comment. **Fig 1b** is only used to suggest the approximate concept of the complementary harvesting behavior. Such behavior can be observed at different time scales, as demonstrated in **Fig. 5** and **6**.

Based on this useful suggestion, we revised **Fig. 1b** by adding the proposed approximate time scale.

Comment 10

What is the SEBS abbreviation, should be discussed near to first mention.

We thank the reviewer for this helpful notice.

We have updated the manuscript on page 4 to include the full name of SEBS (polystyrene-polyethylene-butylene-polystyrene triblock copolymer).

Response:

Comment 11

Fig 2b too small/unclear

Response:

We thank the reviewer for the comments. **Fig. 2b** has been updated to clarify the charge generation mechanism of the textile-based TEG module.

Fig. 2b The charge generation mechanism of the TEG module under in-plane friction between the mover and the stator.

Comment 12

What is the mechanical performance of the devices during folding? The authors report before and after folding and crumpling cycles but not during? Why were such experiments not performed as the more realistic way to test in vivo operational performance?

Response:

We thank the reviewer for the comments about this practical issue. In general, the TEG module is not intended to be used when it is wrinkled, as the contact between the mover and the stator cannot be established for friction-based energy generation. According to the normal charge generation mechanism, substantial performance can be generated under fully contact-separation at the interfaces between two layers. Otherwise, it will bring about a large charge loss when attached to a shaped surface. Many works have tried to address this issue through structural innovation of the module, but have yet to be adapted to the wearable form factors. Future work on developing a highly deformable mechanical energy harvester that can operate electricity during motions will be conducted. Similarly, the BFCs are expected to have optimal performance when it is in good contact with the skin to collect sweat. During folding, due to lack of contact with the skin will limit the performance of the BFC. Thus, in the design, both the BFC and the SC have been strategically placed on the chest area to minimize the bending of the device while ensuring BFCs' contact with the skin. On-body experiments involving exercise have been conducted, as shown in Fig. 5 and 6, where reasonable bending or folding will occur due to the constant motions and frictions. When establishing a secure connection, we did not observe a significant change in performance in the BFCs, TEGs, or the SCs. The experiments were performed only in an in-vivo setting as we aim to characterize the system in a quantifiable fashion with controlled variables.

We have included additional discussions in the manuscript on page 4 to address this comment.

Comment 13

0.833 Hz to 3 Hz is the swinging speed of human arms, ref?

Response:

We thank the reviewer for this useful question. The frequencies used in this study were selected upon estimating the normal arm swinging speed during walking and running. We have performed the output characteristic of TEG under the range of natural arm frequencies of walking (0.81 Hz) and running (2.8 Hz) ergonomically in the literature related biology. [1-2]

We addressed this point on page 7 of the revised manuscript.

[1] J. Nilsson and A. Thorstensson, Adaptability in frequency and amplitude of leg movements during human locomotion at different speeds *Acta Physiol Scand*, 1987, 129, 107-114

[2] G.A. Cavagna, M. Mantovani, P.A. Willems, G. Much, The resonant step frequency in human running, *Eur. J. Physiol.* (1997) 434:678–684.

Comment 14

Fig 3b. CNT-LOx should be CNT-NQ-LOx?

Response:

We thank the reviewer for this careful comment.

Accordingly, **Fig. 3b** has been updated to include the NQ mediator component.

Fig. 3b. A zoomed-in photo image of the fabricated BFC module. Scale bar, 5 mm.

Comment 15

BOD would be better as BOx (LOx is used not LOD in the article) for consistency

Response:

We would like to thank the reviewer for this thoughtful suggestion. Both BOD and BOx has been conventionally used as the abbreviation for bilirubin oxidase. Per your suggestion, BOD has been updated as BOx in the figures and manuscript.

Comment 16

Fig 3. Difficult to read figures e.g. d and f due to similar green line curves. Labels or colours need to be improved for clarity.

Response:

We thank the reviewer for this helpful suggestion.

The labels in **Fig. 3d** and **3f** were updated for easier identification and differentiation.

Comment 17

The testing solution, pH and temperature values should be included in Figures. I understand that it may have always been 0.1M phosphate buffer pH 7.4. for in vitro experiments (capacitance, voltammetry)? If so, this can be repeated a few times or PB or PBS used periodically in the text/figure captions. Page 8. More scientific details for the CNT based biopellet preparation/composition in the text would be appreciated/valuable, as an integral part of the biofuel cell.

Response:

We thank the reviewer for these useful suggestions. The detailed preparation for the pellets was listed in the Methods section and **Supplementary Notes 2**.

The manuscript has thus been revised on page 10 to direct the reader to the experimental details.

Comment 18

Page 9. The BFC was stored in the presence or absence of lactate?

Response:

We thank the reviewer for this question. The BFC was stored dry in the absence of lactate or any other solution.

Comment 19

Page 11. Which area, geometric?

Response:

We thank the reviewer for this useful question. The area in this work was referring to the geometric area of the design.

We have updated the manuscript on page 12 to clarify this point.

Comment 20

How much volume is exactly from the sweat using the PVA-based PBS hydrogel and what is the thickness? Important and interesting details to include in the main publication.

Response:

We would like to thank the reviewer for this critical comment. We have included the thickness and fabrication for the PVA hydrogel used in this work for the BFC module in the manuscript on pages 26-27.

Comment 21

Device powering was continuously demonstrated for 30 min. Is it possible to power for longer or using the same device on different days?

Response:

We thank the reviewer for the comments. The wearable multi-modular e-textile is so durable that it can enough power for much longer and even at least a week. Especially, the stability of a module of biofuel cell has been tested throughout a week with no visible drop in power. Also, the module of TEG can show no optical drop of voltage in the *in-vitro* durability test against sliding frictions for 2000 cycles. We performed an additional washing test (**Fig. S21**) to demonstrate the durability of the microgrid system during extended usage.

Fig S21. Component reliability test under washing tests. (a) Photo demonstrating the washing process of the integrated E-textile system. Room-temperature tap water without detergent was used to wash the shirt using a commercial washing machine for 20 min and dried under ambient environment. (b) The voltage output of the TEG module before and after the washing test. (c) The

power output of the BFC module before and after the washing test in 0 mM and 20 mM lactate in PBS. (d) The GCD of the SC module before and after the washing test.

Comment 22

Methods. What are the molecular weights of all polymers used, and enzyme activities for BOx and LOx?

Response:

We would like to thank the reviewer for the comment to allow us to include any crucial missing chemical information in the method section. Not all molecular weights of the polymers are available from the manufacturer's SDS or CoA. The LOx used in this work is 80 units/mg, and the BOx used in this work is 1.2 units/mg.

Additional information on the material used was included in the revised Methods section on page 24 to help locate the exact chemical.

Comment 23

Methods. For BFC module, how long for the GA crosslinking reaction before drying the sample? Why this time? This detail seems to be missing.

Response:

We would like to thank the reviewer for this useful comment. The GA crosslinking was performed by drop-casting the GA solution onto the pellet and allow the solvent to naturally dry in an ambient environment for 5 min before drop-casting the chitosan layer. The cell was left in the refrigerator at 4 °C overnight before using it. This protocol was adapted from previously published work and has also be discussed in **Supplementary Notes 2-1**.^[1] The details have been added to the Methods section on page 26.

[1] Energy Environ. Sci., 2017,10, 1581-1589.

Comment 24

Methods. When dropcasting LOx, why was BSA and why chitosan at the anode and nafion at the cathode?

Response:

We would like to thank the reviewer for this question. The use of BSA was used as a stabilizer for the enzyme during storage in solution. The chitosan was used on the anode to mechanically immobilize the enzyme to prevent its leaching from the pellets. Similarly, Nafion was used on the cathode to mechanically immobilize the enzyme to the pellets.

We addressed this point on page 26 of the revised manuscript.

Comment 25

SI -the authors used 1 M KCl for half cell characterization of bioelectrodes? Is this not a harsh environment for the enzymes leading to destabilization and deactivation?

Response:

We thank the reviewer for this comment. As mentioned in **Supplementary Note 2-2** (SI, page 7), all the in-vitro characterization for the BFC was performed in 0.5 M PBS with the pH of 7.4.

Comment 26

Figure S10. It is not obvious that there is catalytic oxygen reduction?

Response:

We would like to thank the reviewer for this careful observation. In **Fig. S10 b**, no change should be observed in the current between 0 mM and 20 mM lactate as the cathode ORR reaction is not related to the lactate concentration in the solution. The scale for **Fig. S10 b** has been adjusted to match the ones of **Fig. S10 a**, for the reader to compare the current levels between the anode pellet and the cathode pellet. A zoomed-in view of **Fig. S10 b**, attached for your reference, clearly illustrates that the changes in lactate concentration do not affect the cathode response.

Response to Reviewer #2

General Comment:

In this manuscript entitled authors have discussed the integration of textile based microgrid systems for energy efficient, sustainable, and autonomous wearables. The fabrication of TEG, BFC and SC are discussed in detail and their integration for self-powered applications are also demonstrated by powering wearable sensors.

Overall, I find this work as interesting with sufficient details related to scientific and technological aspects. As far the integration of self-sustained e-textile based bioenergy system is concerned, the work has merits and can be considered for publication after minor revision by addressing the following comments.

Some findings are given below:

Response:

We would like to thank the reviewer for recognizing the quality and attractiveness of our work.

Comment 1

In figure 2 caption and following discussion authors referring capacitor instead of supercapacitor. To avoid confusion please correct it to SC.

Response:

We thank the reviewer for the careful observation that would help clarify our characterization processes. As the characterization for the printed SCs had not yet been conducted in **Figure 2**, we have been using commercial electrolytic capacitors with known capacitance instead of printed SCs for gauging the performance of TEGs.

To clarify this useful point and avoid confusion, the manuscript has been updated on page 7 to clarify this point.

Comment 2

In BFCs characterization authors varied the lactate concentration and measured the performance of the device by analysing the power output. However, in real human sweat apart from lactate other ions like sodium, potassium, urea, amino acids etc are present. Do these ions have any influence on the BFC performance? Did author check the performance of BFC device with different conc. of these ions (with sweat equivalent solution)?

Response:

We would like to thank the reviewer for this insightful question. To address this comment on the performance of the BFC in real human sweat with more complex environments that resemble the constitution of sweats, we have tested the performance of the same BFC module in both PBS and in an artificial sweat that composed of 85 mM of NaCl, 13 mM of KCl and 16 mM of urea in 0.1 M PBS, with different of lactate (0 mM and 20 mM). As shown in the newly added **Fig. S15**, the BFC does not exhibit any significant change in the power output with the addition of sodium, chloride, and urea. There are also other components in the sweat in low concentration, but we do not expect to see them imposing a significant adverse influence on the performance of the BFC. As shown in **Fig. 5c-d** and **Fig. 6e-f** where real, on-body tests with the BFC come in contact with real human sweat, the BFCs were able to power the electronics as expected.

We also have addressed this point on page 10 of the revised manuscript and on page 9 of the Supporting information (**Supplementary Note 2-2**).

Fig. S15. The performance of the BFC module in (a) 0.5 M PBS and (b) artificial sweat that composed of 85 mM of NaCl, 13 mM of KCl and 16 mM of urea in 0.1 M PBS, with the lactate concentration of 0 mM and 20 mM.

Comment 3

In SC fabrication authors used sulfuric acid as the gel electrolyte. However, for wearable application is concerned, the strong acid-based electrolyte may cause some user discomfort. Please comment on it?

Response:

We would like to thank the reviewer for this helpful comment. We are aware that potential leakage of the acidic electrolyte may pose a potential hazard to the user of such wearable systems, and we have adapted engineering solutions to avoid the occurrence of such situations. As mentioned in the Methods section (page 27), a hydrophobic, water-proof SEBS polymer was firstly printed onto the substrate prior to printing the SC modules, and after the gel electrolyte solidify, an additional layer of SEBS polymer was printed onto the entire SC modules to enclose and seal the gel electrolyte, hence preventing its leakage during exercise where sweat may come in contact with the device. We have also added a washing test to the SC (**Fig. S21**, also see below after comment 5), where the SC modules were washed with room temperature water for 20 min and dried. The SC showed no degradation of performance after the washing, which

suggests that no electrolyte leakage took place during even the washing process where complete soaking of water and vigorous stirring was used.

We addressed this point on pages 12 and 27 of the revised manuscript and **Supplementary Notes 3-2** (page 9).

Comment 4

In Figure 5 (a) iii and vi, the TEG device shows a constant voltage output (instead of decreasing) in lactate only region (without sliding). what is the reason for maintaining this constant voltage for TEG device even without any applied mechanical (sliding) force?

Response:

We thank the reviewer for the critical comment. After closely inspect the data and discussion with the persons involved in the characterization, we noticed that during the charging of the capacitor in **Fig. 5a-iii** and **vi**, the oscilloscope was used to record the voltage of the capacitor, whereas, in **Fig. 5a-ix**, the potentiostat was used to record the voltage. Because the oscilloscope has a much larger system impedance compare to the potentiostat, the discharge of the capacitor due to its connection to the instrument was less pronounced in **Fig. 5a-iii** and **vi**. To present the data with better consistency, we have re-tested the system using all potentiostats and updated the plot in **Fig. 5a** accordingly.

Comment 5

Did authors check the performance reliability of the whole micro-grid integrated cloth (having TEG, BFC and SC) under different washing conditions. Since coming to the real time application the stability of the device against washing cycles may be a matter.

Response:

We would like to thank the reviewer for this helpful comment. To fully address this concern in the reliability of the E-textile system under washing conditions, we have conducted a washing test for the system. The shirt containing the BFC, TEG, and the SC was put through a 20-min, room-temperature washing test with normal tap water using a small commercial washing machine, followed by natural drying under the ambient environment. The data were demonstrated in the supplementary material as **Fig. S21**. The output of all three components was tested after washing, and no significant change in the module performance was observed. We do expect the use of hot water, detergent, or hot-air drying may limit the durability of the devices, specifically of the BFC where enzymes were used. Design considerations can be included, where the individual modules can be fabricated into detachable patches that can be taken off, cleaned, and stored in appropriate conditions to further address such concerns.

We addressed this useful point by including additional discussion on pages 8, 10, 12, and 22 of the revised manuscript and on pages 4, 9, and 12 of the Supporting information.

Fig S21. Component reliability test under washing tests. (a) Photo demonstrating the washing process of the integrated E-textile system. Room-temperature tap water without detergent was used to wash the shirt using a commercial washing machine for 20 min and dried under ambient environment. (b) The voltage output of the TEG module before and after the washing test. (c) The power output of the BFC module before and after the washing test in 0 mM and 20 mM lactate in PBS. (d) The GCD of the SC module before and after the washing test.

Reviewer #3:

General Comment:

In this manuscript, the authors attempted to develop a self-powered, multi-functional micro-grid system. Triboelectric generators and biofuel cells were utilized as the energy harvesting module, and the supercapacitors were selected as the energy storage module for the construction of the wearable E-textile. They have also conducted many experiments to prove the complementary, commensurate and compatible features of each functional module. The manuscript is clearly structured and well written. Below are some suggestions for the authors to further increase the manuscript quality.

Response:

We would first like to thank the reviewer for the positive comment on the significance and quality of our work.

Comment 1

The idea is interesting, and I would like to suggest the authors to more highlight the advantage of this design compared with the previous reports. Some related information may be found here, Angew. Chem. 10.1002/ANIE.201207023; Adv. Mater. 10.1002/ADMA.201302951.

Response:

We would like to thank the reviewer for this useful suggestion. Accordingly, we have included both references and related discussion in the introduction on page 2.

Comment 2

Combining the TEG and BFC modules to efficiently scavenge energy form human activities is nice. While would these two modules influence each other in realistic scenarios? For instance, the BFC module worked with the existence of a lot of sweat in the textile, would this affect the triboelectric performance of the TEG?

Response:

We thank the reviewer for the critical comments. As the reviewer mentioned, a sort of sweat or humid air can indeed affect the decrease in the performance of the TEG module. To address this issue, waterproofing, hydrophobic polymer layers were carefully designed, and fast-drying textile was selected in the fabrication of the TEG module to minimize the effect of moisture for reducing the negative effects of humid air on charge screening to the TEG. As a result, no significant decrease in performance was observed during the exercise. An additional experiment of washing the TEG module was also added in **Fig. S21** to demonstrate that the extended exposure to water does not cause any decrease in TEG performance upon drying.

Accordingly, the manuscript has been revised on pages 8 to address this comment.

Fig. S21 (b) The voltage output of the TEG module before and after the washing test.

Comment 3

Most of the data for the BFC were recorded in the lactate solutions. While in realistic application, many factors including the temperature, pH and surface contamination, could significantly affect the BFC behavior, which may be helpful to be considered.

Response:

We would like to thank the reviewer for this insightful comment. Indeed, the performance of the BFC can be affected by many factors, such as lactate concentration, pH of sweat, and temperatures. Such behavior has been studied extensively in previous works, showing extended exposure to low pH or high temperature can cause a decrease in the LOx enzyme activity.[1-2] We have added an experiment (Fig. S15) where artificial sweat including a high concentration of sodium, potassium, chloride ions, and urea instead of only PBS was used in characterizing the performance of the BFC, which did not show any significant change in the BFC performance.

Page 10 of the revised manuscript and page 9 of the revised Supplementary information has been updated to address this issue.

Fig. S15. The performance of the BFC module in (a) 0.5 M PBS and (b) artificial sweat that composed of 85 mM of NaCl, 13 mM of KCl, and 16 mM of urea in 0.1 M PBS, with the lactate concentration of 0 mM and 20 mM.

[1] Chem. Comm. 2020, 56, 2004-2007

[2] Adv. Funct. Mater. 2020, 30, 1906243

Comment 4

The authors claimed that the capacitance of each SC was $\sim 150 \mu\text{F}$, which may be limited to power various electronics. Will it be possible to integrate flexible batteries with high capacities into this E-textile system?

Response:

We thank the reviewer for this helpful suggestion. In the current system, the capacitance of each unit within one SC module has the capacitance of $\sim 750 \mu\text{F}$ and would add up to $\sim 3.75 \mu\text{F}$ per module if each unit were connected in parallel instead of in series. As we intend to increase the voltage threshold of each SC module, their capacitance values appear to be very low. Additionally, in this work, we have accurately calculated the amount of capacitance needed for powering the electronics, as an extra amount of capacitance would result in the undesirable long system booting time, and a lower amount of capacitance would not be sufficient for powering the electronics. Certainly, flexible batteries can be integrated into the E-textile system to supply power for electronics with higher power and energy demand. However, in parallel, the power density of the energy harvesters has to be improved extensively to match the energy rating of such batteries. The current μW -level harvesters would require hours or up to days to charge a battery with even a moderate amount of capacity and does not align with the “commensurate energy rating” design concept proposed in this work. Future work will be conducted to explore the possibility of integrating energy storage devices with different capacity and power ratings for different applications.

We addressed this important point on pages 22-23 of the revised manuscript.

Comment 5

The BFC and TEG modules can only collect ambient mechanical energy at sports mode while most people need to stay quietly to work. I wonder about the possibility of an energy harvesting system that requires lower dependency to environment and usage scenario. Some comments may be helpful to the readers.

Response:

We would like to thank the reviewer for this valuable comment. Indeed, the possibility of developing bioenergy harvesters with less dependency on both external environmental factors and high-intensity movement is of extreme interest for an autonomous, self-powered system. Currently, microneedle-based, ingestible, or implanted BFCs that are powered by the metabolites inside the body have been explored. BFC integrated onto contact lenses that use the metabolite in the tears were also been studied. Such systems do not require the constant input of movements. Mechanical energy harvester or self-powered sensors that can be powered by breath, pulses, or heartbeats were also explored. Such innovative approaches are still somewhat limited by their invasiveness for wearable applications but may be extremely desirable for developing self-powered implanted or ingestible systems for in-body sensing and monitoring. We envision the

effective, non-invasive, and passive extraction of biofuels to be a point of a breakthrough for various BFC harvesters, and the continued exploitation in the passive movements on our body will still be of great interest in the coming decade for the biomechanical energy harvesting.

We addressed and clarified this important point on pages 22-23 of the revised manuscript.

Reviewer #1 (Remarks to the Author):

The manuscript is very interesting and the improvements made by the authors respond perfectly to my requests and comments. The article is clearly written and structured and can be now accepted for publication:

Reviewer #2 (Remarks to the Author):

Authors have satisfactorily addressed my comments. Looking forward to publication of this paper.

Reviewer #3 (Remarks to the Author):

The authors have addressed my concerns with appropriate revisions, and I will thus recommend acceptance now.